# Remote Psychotherapy during the COVID-19 Pandemic: A Mixed-Methods Study on the Changes Experienced by Austrian Psychotherapists

**DOI:** 10.3390/life13020360

**Published:** 2023-01-29

**Authors:** Michael Stadler, Andrea Jesser, Elke Humer, Barbara Haid, Peter Stippl, Wolfgang Schimböck, Elisabeth Maaß, Helmut Schwanzar, Daniela Leithner, Christoph Pieh, Thomas Probst

**Affiliations:** 1Department for Psychosomatic Medicine and Psychotherapy, University for Continuing Education Krems, 3500 Krems, Austria; 2Austrian Federal Association for Psychotherapy, 1030 Vienna, Austria; 3Österreichische Gesellschaft Für Wissenschaftliche, Klientenzentrierte Psychotherapie und Personorientierte Gesprächsführung (ÖGWG), 4020 Linz, Austria

**Keywords:** remote psychotherapy, psychotherapy via telephone, psychotherapy via videoconferencing, tele-health, e-mental-health, COVID-19, pandemic, psychotherapy, qualitative psychotherapy research, mixed-methods psychotherapy research

## Abstract

The outbreak of the COVID-19 pandemic and associated measures to contain the SARS-CoV-2 coronavirus required a change in treatment format from face-to-face to remote psychotherapy. This study investigated the changes experienced by Austrian therapists when switching to psychotherapy at a distance. A total of 217 therapists participated in an online survey on changes experienced when switching settings. The survey was open from 26 June until 3 September 2020. Several open questions were evaluated using qualitative content analysis. The results show that the setting at a distance was appreciated by the therapists as a possibility to continue therapy even during an exceptional situation. Moreover, remote therapy offered the respondents more flexibility in terms of space and time. Nevertheless, the therapists also reported challenges of remote therapy, such as limited sensory perceptions, technical problems and signs of fatigue. They also described differences in terms of the therapeutic interventions used. There was a great deal of ambivalence in the data regarding the intensity of sessions and the establishment and/or maintenance of a psychotherapeutic relationship. Overall, the study shows that remote psychotherapy seems to have been well accepted by Austrian psychotherapists in many settings and can offer benefits. Clinical studies are also necessary to investigate in which contexts and for which patient groups the remote setting is suitable and where it is potentially contraindicated.

## 1. Introduction

The outbreak of the COVID-19 pandemic necessitated a sharp modification in the psychotherapeutic treatment format [1]. To contain the spread of the SARS-CoV-2 coronavirus, restrictive measures such as restrictions on outdoor activities, quarantine and social distancing were implemented. Psychotherapists and patients alike were faced with the challenge of adapting to a new, virtual setting within a very short time [2,3]. The number of patients treated psychotherapeutically via telephone or the internet increased sharply. During the first curfew in 2020, the number of patients treated via telephone on average per week in Austria increased by 979% and via the internet by 1561% (both *p* < 0.001) [4]. Study results confirmed that the infectious disease COVID-19 has a clear impact on the practice of psychotherapy in Austria [3]. 

In Austria, psychotherapy via digital media was not considered lege artis until that time [5] and was also not covered by health funds. The legal regulations changed with the outbreak of the COVID-19 pandemic and the associated need to restrict socio-physical contact [6]. This is not the least since study results indicate that psychotherapy via the internet can be regarded as equally effective to psychotherapy in face-to-face contact [7,8]. There seems to be no significant difference in effectiveness between the treatment modes of face-to-face setting, real-time video conferencing and telephone [9]. Psychotherapy via videoconferencing has already been described as promising and tends to be equivalent in patients with anxiety disorders [10]. In a study by Bouchard et al. [11] involving 71 patients with panic disorder and agoraphobia, a very strong therapeutic alliance was demonstrated in video conferencing, which did not differ from treatments in face-to-face settings. Similarly, research findings suggest the effectiveness of treatment via telephone for depressive symptoms [12,13,14,15]. Moreover, dropout rates might be lower for telephone therapies than for treatments in face-to-face settings [16].

While there was still a general skepticism among therapists towards psychotherapy at a distance before the COVID-19 pandemic, despite the positive results of efficacy research [17,18], this attitude has since changed due to experiences gained during the first months of the pandemic [19,20]. While the digital treatment setting did not play a substantial role in healthcare before COVID-19, the outbreak of the COVID-19 pandemic also pushed former skeptics to work at a distance [21]; it seems that quite often, the experiences made in the process have led to a re-evaluation of remote psychotherapy [20]. At the same time, treatment at a distance is not considered by therapists to be fully comparable to the face-to-face setting [19].

In the remote setting, the therapeutic alliance between patients and therapists was described by therapists as impaired, for example, due to the loss of the physical exchange or the lack of various sensory impressions, and was experienced as more superficial and businesslike. Moreover, therapists were confronted with technical challenges and security-related issues regarding the safeguarding of confidentiality in the online setting [22]. Furthermore, therapists, psychologists, and social workers also expressed concerns about building and maintaining the therapeutic alliance [23]. Therapists seem to perceive greater differences between treatment settings in face-to-face contact and at a distance than patients [24]. Moreover, therapists’ satisfaction with psychotherapy via videoconferencing seems to be related to their level of professional maturity and experience, as older therapists with previous experience in the video-based setting have a more positive attitude towards remote psychotherapy [25]. Furthermore, study results indicate that therapeutic interventions also differ between settings, and interventions of different psychotherapeutic orientations are more distinct in face-to-face contact than in remote psychotherapy [26].

## 2. Materials and Methods

### 2.1. Study Design and Procedure

The primary objective of the current study was to collect data on the changes experienced by Austrian therapists when switching from face-to-face to remote psychotherapy and/or from remote to face-to-face psychotherapy during the first year of the COVID-19 pandemic. Specifically, we wanted to find whether psychotherapists experienced changes in the therapeutic relationship (research question 1), whether they experienced changes in the content of the sessions (research question 2), whether they noticed changes in the intensity of the sessions (research question 3), whether the structure of the sessions changed in their practice (research question 4), and how they experienced the digital setting and the lack of physical presence (research question 5). A further aim was to investigate potential differences in these experiences with respect to the sociodemographic and professional characteristics of psychotherapists.

Following an exploratory research approach, we conducted a cross-sectional online survey among Austrian psychotherapists authorized to provide psychotherapeutic treatment to patients. This includes psychotherapists in training in recognized training institutions, who are already working under supervision after the fourth year of training and psychotherapists registered at the Austrian Federal Ministry of Social Affairs, Health, Care, and Consumer Protection. Registered psychotherapists have completed their training as psychotherapists and are officially licensed to treat patients. Health insurance companies only refund psychotherapy sessions provided by registered therapists. The survey was set up with REDCap (Research Electronic Data Capture) [27,28] and was open from 26 June 2020 until 3 September 2020. By this time, the COVID-19 measures imposed by the government had been largely relaxed. A previous initial lockdown in Austria from March 16 to 30 April 2020, mandated that Austrians were only allowed to leave their homes for certain activities, such as covering important basic needs, caring for others in need or going to work. During this time, many therapists switched from face-to-face psychotherapy to remote sessions [4].

The survey included a total of 128 questions covering basic sociodemographic variables; the number of patients who were switched from face-to-face to remote psychotherapy or from remote to face-to-face psychotherapy; therapists’ experiences with digital media; the type of media used; an assessment of the various therapeutic interventions used in the different settings; as well as several open-ended questions about the therapeutic relationship, content, intensity, and structure of remote as compared to face-to-face sessions and therapists’ experience of the lack of physical presence and the spatial distance between themselves and the patient. Open-ended questions were formulated to elicit as wide a range of perceptions as possible. We estimated that it would take psychotherapists 15–20 min to complete the questionnaire in its entirety. In the present study, only the open-ended questions from the survey were analyzed. Quantitative findings are published elsewhere [26].

The study received approval from the ethics committee and the data protection officer of the University for Continuing Education Krems (EK GZ 27/2018-202). We follow the APA Journal Article Reporting Standards for Qualitative Research in Psychology [29] in the presentation of our research.

### 2.2. Participants

Austria has a long tradition of psychotherapy and a wide range of 23 accredited psychotherapy schools [30]. They can be classified into four orientations. The largest orientation is the humanistic orientation (37.8% of the psychotherapists in Austria), followed by the psychodynamic orientation (25.9% of the psychotherapists in Austria), the systemic orientation (24.3% of the psychotherapists in Austria) and the behavioral orientation (12.0% of the psychotherapists in Austria).

A link to the online survey was sent to all registered psychotherapists by the last author in cooperation with the Austrian Federal Association for Psychotherapy, which supported the study. Continuing education credits points were awarded as an incentive for participation. In addition, psychotherapists in training who were already treating patients under supervision were invited to participate in the survey. Email lists for psychotherapists in training were provided by the Austrian Federal Association for Psychotherapy. In addition, a link was sent to psychotherapy students from the University for Continuing Education Krems, which is one of several institutions offering training as a psychotherapist. In total, *n* = 222 respondents participated in the survey. All participants gave electronic informed consent after reading the data protection declaration. Five therapists did not experience a change of treatment format and were therefore excluded from further analyses, resulting in a final sample of *n* = 217 therapists.

### 2.3. Measures

The study comprised 10 open-ended questions. 

Q1: In your own words, please describe how the therapeutic relationship with your patients changed as a result of the switch from psychotherapies in personal contact to psychotherapies via digital media.Q2: In your own words, please describe how the content of sessions changed as a result of the switch from psychotherapies in personal contact to psychotherapies via digital media.Q3: In your own words, please describe how the intensity of sessions changed as a result of the switch from psychotherapies in personal contact to psychotherapies via digital media.Q4: In your own words, please describe how the structure of sessions changed as a result of the switch from psychotherapies in personal contact to psychotherapies via digital media.Q5: How do you experience the lack of physical presence in remote psychotherapy sessions?Q6: How do you experience the spatial distance and remaining in your own space (not in the therapy room) when you conduct psychotherapy via digital media?

Questions 1–4 were also asked regarding the switch back from the remote setting to face-to-face psychotherapy. As many psychotherapists referred to remote psychotherapy in their responses, these responses were also included in the analysis and coded together with the respective question addressing the switch from face-to-face to remote psychotherapy.

### 2.4. Data Analysis

We used a conventional approach to qualitative content analysis [31]. In conventional content analysis, categories are derived from the data rather than from theory. It is generally applied in study designs that aim to describe a phenomenon about which little theory or literature is yet available.

Out of 217 respondents, 63 answered all open-ended questions, 143 answered at least one, and 11 did not fill in any free text field. Many of the answers were very detailed. We received only a few keyword-like responses, as is usually the case with open-ended survey responses. In sum, we received 1448 free text comments: 308 describing changes in the therapeutic relationship, 275 describing changes regarding the content of sessions, 265 describing changes in intensity, 238 describing changes in the structure of sessions, 192 focusing on experiences regarding the lack of physical presence in remote psychotherapy and 170 addressing the spatial distance. Overall, responses to Q1 were the most comprehensive. Respondents addressed various aspects of their own accord that not only had to do with the therapeutic relationship but also related to the subsequent questions. As a result, some answers were repeated in later questions. They were only coded if new aspects were addressed.

At the beginning of the coding process, two coders read through the whole data set to familiarize themselves with the material. Subsequently, data were imported into Atlas.ti for coding [32], and one coder read through the material again, inductively defining categories in the process. After coding 30% of the material, the second coder coded the same material with the list of categories and category definitions provided by the first coder. To enhance reliability [33], we assessed the agreement of how the two coders coded the data set [34]. Percentage agreement was high at 94.3%, and inter-coder agreement using Krippendorff c-α-binary = 0.985. Any citation on which the coders disagreed was discussed between the two coders, and the category definitions were expanded in this process. In addition, this step of the coding process created larger thematic clusters to which categories were assigned. Afterward, the second coder coded the entire data set, documenting the cases in which assignment to a category was not clear. These cases were coded together.

Chi-squared tests were conducted to analyze potential differences in the frequency of main categories reported by psychotherapists in terms of sociodemographic (years of age: ≤40, 41–50, 51–60, >60; gender: female, male) and professional characteristics (years in the profession: ≤5, 6–10, >10; psychotherapeutic orientation: psychodynamic, humanistic, systemic, behavioral). Differences in the frequencies of subcategories were only analyzed with respect to gender, as the number of coded text passages in the subgroups of the different orientations, age and experience groups was insufficient to make reliable inferences about the population of psychotherapists. To analyze differences in the length of the free text answers, *t*-tests (gender) and univariate ANOVAs (age group, professional experience group, psychotherapeutic orientation) were applied. Statistical analyses were performed in SPSS version 26 (IBM Corp, Armonk, NY, USA). *p*-values of ≤0.05 were considered statistically significant (2-sided tests).

## 3. Results

### 3.1. Sample Description

A total of 77% of respondents were female, and 23.0% were male. They were M = 50.66 (SD = 9.65) years old, and while most of them were certified psychotherapists in Austria (91.2%), 8.8.% worked under supervision in the last part of their training to become psychotherapists. Regarding their professional experience, M was 10.61 years (SD = 9.50) (value was set to 0 for psychotherapists working under supervision). Overall, 46.1% of respondents practiced humanistic psychotherapy, 22.6% practiced psychodynamic psychotherapy, 20.7% belonged to the systemic orientation and 10.6% to the behavioural orientation (10.6%).

Most respondents worked in their private practice (96.8%), and 39.6% had gathered experience with remote psychotherapy before COVID-19. Most psychotherapists used psychotherapy via telephone (88.5%) or videoconferencing (76.5%). Psychotherapy via email was used by 22.6% of respondents, and 9.2% used chats or other digital media (2.8%). M = 11.21 (SD = 10.12) patients were switched from face-to-face psychotherapy to remote psychotherapy, and M = 9.62 (SD = 10.34) patients were switched from remote to face-to-face psychotherapy.

### 3.2. Results of Qualitative Analysis

The analysis resulted in seven main categories, which each comprise several subcategories. Figure 1 represents the main categories and their frequency in relation to the number of respondents. We chose not to report the number of coded text passages but always reported the number of respondents per category. Some respondents commented more often on a topic. These responses are reported descriptively but are not reflected in the frequencies. The order of the categories is not based on the frequency of the categories but was chosen so that the presentation of the contents of the subcategories builds on each other as coherently as possible.

Differences in the frequencies of two main categories emerged between female and male psychotherapists (Table 1): female psychotherapists reported difficulties more often (38%) compared to their male colleagues (20%; *p* = 0.02). Additionally, more female psychotherapists expressed experiences related to the therapeutic relationship (79% vs. 52%; *p* < 0.001).

Frequencies of all investigated main categories did not differ among age groups (*p* ≥ 0.08), groups of professional experience (*p* ≥ 009) and among the four therapeutic orientations (*p* ≥ 0.06). 

Female psychotherapists provided longer comments (mean number of characters (M) = 932.6, SD = 751.0) than male psychotherapists (M = 595.8, SD = 630.1), *t* (94.5) = 3.116; *p* = 0.002. The length of the text answers was neither associated with the age, the professional experience, nor the therapeutic orientation of the participating psychotherapists (*p* ≥ 0.11).

#### 3.2.1. Benefits

A total of *n* = 56 respondents (25.8%) reported the benefits of remote psychotherapy. Figure 2 displays the allocated subcategories. 

*N* = 6 (2.8%) commented positively that therapies could also be provided in the case of illness. For high-risk patients and pregnant patients, the risk of infection through travel and face-to-face contact in the practice was eliminated. An important subcategory, mentioned by *n* = 39 (18%) respondents, addresses the flexibility enabled by remote psychotherapy. Respondents reported that they were able to offer appointments more flexibly than usual, even at short notice. This proved beneficial in crises or for patients who needed a higher frequency of sessions. The possibility of fitting sessions in between appointments proved helpful for psychotherapists and patients in scheduling sessions, as did the fact that there was no need to travel to and from the practice. Mothers of younger children, in particular, benefited from increased flexibility. Spatial independence ensured that business trips, study abroad, and even vacations were no longer an obstacle to offering or attending psychotherapy. Other *n* = 20 (9.2%) respondents observed remote sessions to be more comfortable both for themselves and their patients. Clothing, food, drink, and not having to go out in bad weather were mentioned, as well as being unobserved on the phone. *N* = 6 (2.8%) mentioned other advantages, for example, having more resources available at home, such as books for consultation, or being able to maintain a professional distance more easily. 

Frequencies of all investigated subcategories did not differ between female and male psychotherapists (*p* ≥ 0.112).

#### 3.2.2. Drawbacks

*N* = 106 respondents (48.8%) reported the drawbacks of remote psychotherapy. Subcategories are displayed in Figure 3.

*N* = 31 (14.3%) respondents reported technical problems: connection problems, delays or interruptions in the transmission of sound and images and poor sound and image quality. These interfered with therapy processes and made it difficult for therapists to tune in to their patients. Therapists were also required to ensure the functionality of the technology and to comply with data protection regulations, which was an additional challenge for some.

Other (*n* = 64 (29.5%)) respondents noted that remote psychotherapy made them feel more exhausted. They reported fatigue from longer screen time and distractions at home. They also described that it required more concentration (1) to compensate for the lack of perceptions and capture patients’ emotions (passive) and (2) to convey empathy through verbal communication only, in case of sessions on the phone (active). Respondent 170 mentioned how *“it was more exhausting to find the “right words” because all other sensory channels were eliminated”.* Additionally, respondent 172 commented that *“Over the phone, it was difficult and required a lot of concentration to capture emotions only through the spoken word”*.

In this context, *n* = 26 (12%) respondents observed that they or their patients were more distracted in the remote setting. Other people in the household and pets were described as distractions for both therapists and patients. Therapists perceived patients as distracted if they ate or drank during psychotherapy sessions. They experienced themselves as distracted by glimpses into the patients’ private spaces, by the environment at home or by seeing their own faces on the screen during videoconferencing sessions. *N* = 27 (12.4%) respondents, who worked from home, reported that it was challenging to maintain a separation between their personal and professional life. It was more difficult for them to distance themselves from work, pay attention to a healthy work-life balance, and maintain clear work schedules. It was also harder to adopt a therapeutic stance.

No differences between female and male psychotherapists in the frequencies of mentioned drawbacks were observed (*p* ≥ 0.372).

#### 3.2.3. Difficulties

*N* = 74 (34.1%) respondents addressed the difficulties encountered in remote psychotherapies. Figure 4 illustrates the subcategories.

*N* = 14 (6.5%) mentioned having worked remotely only with patients who had been in therapy for some time. First consultations were described as difficult. In particular, establishing a holding psychotherapeutic relationship was mentioned as a challenge. *Noncommittal* (respondent 79), *businesslike* (respondent 151), and *insecure* (respondent 208) were adjectives therapists used to describe their experience.

Other *n* = 37 (17.1%) respondents described difficulties in upholding the therapeutic setting. The temporal frame was an issue for some. Therapists reported they had to contact patients more often, for example, to remind them of sessions. For other respondents, the focus was on the therapeutic space. They expressed a sense of losing control in the remote setting. As respondent 193 expressed, *“The setting was eroding”.* Although it was possible for therapists to choose the room in which they located themselves, it was not possible to exert any influence on the spatial conditions at the patients’ homes. Disturbances by family members were mentioned very frequently as well as *“undignified conditions”* (respondent 178), e.g., in case patients had to attend a session from their bathroom or from behind a paravan. Patients had to take care of adequate conditions “on their side” themselves, which required more self-responsibility. Therapists also described it as difficult not having any influence on the technical connection or on how patients participated in the session, e.g., lying in bed, eating, etc.

Finally, we generated the category “difficulties or benefits for certain diagnoses and patient groups”, which was addressed by *n* = 40 (18.4%) respondents. Not only difficulties but also benefits were subsumed under this category. Several respondents mentioned difficulties with remote psychotherapies for patients with structural deficits who needed a lot of stabilization. There appears to be little consensus and very contradictory statements regarding the different types of mental disorders. For anxiety disorders and schizophrenia spectrum disorders, staying in one’s own room was described as building confidence and reducing anxiety by some respondents. Other respondents found it more difficult to treat these patients remotely. Some respondents also saw advantages in treating traumatized patients in a remote setting. They described that patients were able to address traumatic events for the first time in the remote setting, as it helped them to distance themselves from the event and reduced feelings of shame. Not being seen during their account had a disinhibiting effect. Other respondents experienced working with traumatized patients as more difficult in the remote setting because patients could not be supported so well, for example, when dissociating. As respondent 151 put it: 


*“Traumatic experiences were addressed. But for me as a therapist, the patient in tears on the phone is an experience that I do not wish to repeat. I had a strong feeling that I could not fulfill my responsibility as a therapist. Even without touching the patient in such situations, I am convinced that my physical presence alone and my staying present are important for the patient. Also, the thought came to my mind: what do I do when he/she throws down the phone and—yes, what does he/she do? jumps out of the window, runs into the street without looking left or right...”*


Another patient group that was mentioned several times was the group of children and adolescents. Therapists found it more difficult to work remotely with children, as the remote setting made it more difficult to engage in playing and relied a lot on verbal communication. There were more positive observations regarding remote psychotherapy with adolescents. It was noted that remote formats are familiar to young people and are therefore a good way to get in touch.

Among the experienced difficulties of remote psychotherapy, gender differences became visible for the subcategory “difficulties or benefits for certain diagnosis and patient groups”, with female psychotherapists reporting more often respective experiences (21%) than male psychotherapists (8%; χ^2^ (1) = 4.383; *p* = 0.036). For the remaining two categories, no differences were observed with respect to gender (*p* ≥ 0.144).

#### 3.2.4. Modifications of the Setting

*N* = 134 (61.8%) respondents made statements regarding how they dealt with changes in the setting and how they adapted to the new situation. Subcategories are shown in Figure 5.

The subcategory “handling the new setting”, mentioned by *n* = 72 (33.2%) respondents, describes statements from respondents about how the move to remote psychotherapy initially caused a sense of uncertainty among therapists and among patients. For the most part, respondents described continuing to hold sessions from their practice rooms. This was felt to help separate work from private life, maintain a professional attitude, and provide continuity for patients in the form of a familiar space. 

*“The beginning was structured by me, when I started the Zoom call and let the (waiting) person in. It was important to me to maintain continuity and stability in the sense that I told and showed the patients that I was sitting in the usual armchair in the practice room. Their familiar space thus continued to exist, only they were not spatially there, I was connected to them via telephone or Zoom”.* (respondent 78)

Alternatively, some therapists switched to their home office. In this case, the background visible on the screen was arranged in such a way as to create a professional context (e.g., removal of personal items and pictures, covering glass doors to other living spaces, etc.). In addition, respondents explained that it was important for them to convey a sense of security to the patients. This also included establishing the new setting, e.g., discussing from where patients participated in the session and whether undisturbed communication was possible in this environment, how to handle the software and deal with technical failures, how data protection regulations were complied with, who could provide help in the event of a crisis and who could also be reached by the therapist, etc. Respondents mentioned how they had to be more demanding that patients ensure adequate setting conditions “on their side” or adhere to setting conditions, such as starting times. As respondent 85 stated: *“Discussing “rules of conduct” in advance is important (e.g., pat. not just hanging up, closing laptop)”*.

In a few cases, respondents described that patients went for a walk during the session because they could not establish an undisturbed atmosphere at home. Respondents also discussed the possible advantages and disadvantages of remote psychotherapy with patients and inquired about patient expectations. Some therapists also mentioned new rituals they introduced for beginning and ending a session via videoconferencing or on the phone.

Another subcategory mentioned by *n* = 44 (20.3%) respondents concerns changes in the frequency and duration of sessions. Some respondents explicitly mentioned not changing anything about the structure of the sessions (day of the week, time, duration). However, another part of the respondents reported more frequent or less frequent, more irregular or more regular and shorter sessions. In addition to videoconferencing and telephone sessions, some therapists also communicated with patients in writing (text messaging, email, chat).

A very large subcategory mentioned by *n* = 77 (35.5%) respondents concerns the use of therapeutic interventions. The majority of respondents described having to forgo many interventions in the remote setting. In particular, interactive interventions (e.g., role-playing, constellation work, etc.), body-based interventions (demonstrations, movement, EMDR, hypnotic trances, autogenous states of relaxation), art therapy interventions (visualizations, sand play, sculptures, etc.), therapeutic play in work with children, animal-assisted interventions, and work with objects or with guided affective imagery were mentioned. In contrast, other interventions were used more frequently, such as the assignment of homework. The therapeutic conversation also gained importance in remote psychotherapy, as much of the content was addressed verbally. Some respondents described that interventions were more difficult to apply but could be adapted for the remote setting. For example, some body-based interventions could be delivered in an adapted form, such as autogenic states of relaxation or hypnotic trances, as could interactive interventions, such as role-playing and guided affective imagery. Some respondents reported using handouts, exercise sheets, and audio that they gave to patients to take home. The overall impression was that respondents used interventions primarily in a stabilizing or resource-strengthening way and focused on techniques that activated cognition. In contrast, they worked in a less confrontational, less regression-promoting, and less emotion-activating manner. Trauma-specific interventions were also used with caution.

Handling the new setting was mentioned by more female (38%) than male (18%) psychotherapists (χ^2^ (1) = 6.752; *p* = 0.009). For the other two subcategories, no differences were observed between male and female therapists (*p* ≥ 0.714).

#### 3.2.5. Lack of Physical Presence

*N* = 146 (67.3%) respondents named categories related to the absence of physical presence. The subcategories are displayed in Figure 6.

*N* = 21 (9.7%) respondents noted that the office was lacking as both a physical and an intrapsychic space in remote psychotherapies. They observed that patients missed the time in the office away from their usual contexts. Also missing was the journey to and from the office as a mental space for reflective engagement with what patients wanted to talk about or had worked through in the session. Sessions took place more *“in-between”* (respondent 65). As one respondent put it: *“Patients reported that it is unusual when the journey home can no longer be experienced and one is back in “real” life from one second to the next. Processing what was discussed suffers”.* (respondent 30) Therapists also missed rituals that had shaped the therapeutic encounter in the office, such as inviting patients in, shaking hands, offering a drink or passing a handkerchief.

A majority (*n* = 118 (54.4%)) of respondents made mention of impaired sensory perception in remote psychotherapies. They referred to the perception of nonverbal communication signals and body language, such as facial expressions, gestures, posture, movements, ideomotor activity, and breathing. They also reported altered acoustics, lack of smell, and eye contact. It was mentioned that it became more difficult to gather diagnostic information, to emotionally tune in to the patient, and to assess the effect of interventions. Respondent 94 commented, *“The distance made it more difficult to perceive, to sense, to observe”.* The difficulty of assessing the atmosphere was mentioned in particular. The assessment of silence was mentioned several times in this context, as here by respondent 59: 


*“A young woman wanted to stay in contact via telephone—in this case it was difficult, especially for me, to assess her reactions without having an image (silence—is she thinking about what has been said or is she crying quietly??? Difficult to assess; asking was disruptive in the process)”*


In addition, body-oriented psychotherapists pointed out the lack of (inter-)bodily perception. Respondent 53 described that *“physical encounters support the process of emotional processing. Traumatic experiences can be better processed through therapeutic physical proximity”.*

*N* = 65 (30%) respondents reported how they tried to replace missing sensory perceptions by focusing on existing sensory channels. They attached particular importance to attentive listening, the perception of speech melody, tonality and subtleties in speech (formulations, choice of language and words, pauses, speaking pace, volume, etc.). Respondent 179 noted, *“the lack of physical presence focused my attention on listening and the words used and was just as intense”.* Other respondents remembered how an imaginary image of the patient was formed during telephone contact. Therapists also observed that they used their voice and speech more consciously to stay in contact with their patients. In the case of video conferencing, respondents described how they paid close attention to what was visible on the screen. Respondent 73 observed, *“only a section of the patient is visible, but you focus on details that are otherwise not present to this extent”.* Respondent 54 recounted:


*“The “large format” of the upper half of the body during video chat, with the visibility of subtle changes in facial expressions, had its own “physical” presence for me. When I was on the phone and the patient’s voice was close to my head, I also experienced a special kind of presence. Even when chatting, I had perceptions of the patients’ bodily presence caused by what they wrote”*


Also visible were glimpses into the private spaces of patients, which were used by many therapists as additional diagnostic information (as described by respondent 159) or to bring themes into therapy (as described by respondent 110). *“I now know a lot more about the patient’s living environment, which was readily opened up to me as well—of high diagnostic relevance!”* (respondent 159). *“The personal environment of the patient was more concrete for me and thus possible to include directly”.* (respondent 110). In addition, respondents described how they obtained missing perceptions by asking for them. For example, respondent 40 noted, *“missing observations were discussed verbally”.* However, this places a high demand on patients to put their perceptions into words, as respondent 195 thematized: *“Patients are extraordinarily challenged in verbalizing their emotions”.*

Male psychotherapists reported more often about the lack of the office as a therapeutic space (18.0%) than female psychotherapists (7.2%; χ^2^ (1) = 5.148; *p* = 0.023). For the other two subcategories, no differences were observed between male and female therapists (*p* ≥ 0.080).

#### 3.2.6. Psychotherapeutic Relationship

A major category mentioned by *n* = 159 (73.3%) respondents is the “psychotherapeutic relationship”. It includes as subcategories various aspects related to the quality and intensity of the therapeutic relationship. The four subcategories are illustrated in Figure 7.

*N* = 98 (45.2%) respondents described that the therapeutic relationship was strengthened or even intensified by the fact that patients experienced that their therapists were there for them even in the crisis and that psychotherapies were continued in the remote setting. Respondents stated that patients were very “*grateful*”, “*happy*”, “*relieved*”, “*unburdened*” or reacted “*positively*”. For example, respondent 28 voiced: 


*“I offered all my clients to use the new forms immediately after the announcement of the ÖBVP (the Austrian Federal Association for Psychotherapy, which informed psychotherapists that sessions were to be held remotely if possible), and this was received with “gratitude” or “relief “. Some were afraid/worried about having to “go through the crisis alone”. The quick provision of alternatives certainly had a positive influence on the relationship (“She doesn’t leave me alone”, “She is also there for me in the general crisis”)”*


Repeatedly, the shared experience of the crisis was considered as uniting, as was the fact that patients, as well as therapists, were in lockdown, attended the sessions from home and sometimes both were navigating (technical) “uncharted territory”. For example, respondent 42 described:


*“Conversations were more personal because of Corona—in the sense that you share the lockdown situation. We are more or less in the same boat, and have similar difficulties (small apartments, bad WiFi, no childcare—so children who “barge in”, etc.)—these are things you simply catch through Corona, because especially in the beginning everything was new, untested, spontaneous, complicated by external circumstances. (…) It was also more personal because of the way of communication: the patient is sitting comfortably at home, with a cup of coffee or tea, in familiar surroundings, without makeup and in her sweatpants, and she is just happy to be able to have contact with someone, due to Corona. This changes the nature of the conversation. I, as a therapist, of course tried to have a professional ambiance, yet I was also at home and in a similar situation”*


In addition, many respondents described how relational closeness and intimacy were generated in remote contact. They used adjectives such as “*open*”, “*confidential*”, “*personal*”, “*holding*”, “*trusting*”, “*strengthening*”, “*reliable*”, “*intense*”, “*consolidated*”, “*stable*”, “*cooperative*”, “*deepened*”, “*connected*”, “*secure*” and “*intimate*” to describe their and their patient‘s relational experience. Sometimes the closeness in remote contact was described as a “special” or “different” kind of closeness than that in face-to-face contact.

In this context, *n* = 23 (10.6%) respondents observed that in the remote setting, the atmosphere was more relaxed, and there was less negative transference in the therapeutic relationship. They attributed this to the spatial separation and to the fact that patients were at home in their safe environment.

In contrast, *n* = 86 (39.6%) respondents mentioned that they experienced less closeness in the psychotherapeutic relationship during remote sessions. “*Superficial*”, “*difficult*”, “*distant*”, “*impersonal*”, “*noncommittal*”, “*flattened*”, “*fragile*”, “*lonely*”, “*alienated*”, “*cold*”, “*businesslike*”, “*less palpable*”, “*less immediate*”, “*foreign*”, “*uncertain*” and “*reserved*” were adjectives used to describe relational experiences in remote sessions. In many cases, this was attributed to the fact that the other person is more difficult to “grasp” emotionally in remote contact and that atmospheric information is lost. Respondent 11 described this as a “*lack of relational immediacy*”, and respondent 30 stated, “*I felt like I couldn’t grasp the patient as well. It was more difficult to assess the client’s emotional situation to the same extent as in a face-to-face conversation”.* Respondent 138, in turn, commented “*on the relationship level, it was no longer possible to “tune in” as usual*”, and respondent 154 elaborated by stating, “*establishing a presence in the relationship, being empathically accurate and empathizing at the moment and being congruent/immediately involved is more difficult, as a result of which the flow of the relationship often falters”.*

In this context, *n* = 33 (15.2%) respondents observed that the holding function is impaired in the remote setting, i.e., respondents see their ability to emotionally support patients in crises, to provide support in difficult situations or to work through difficult issues therapeutically as limited.

Female psychotherapists expressed strengthening of the relationship more frequently than their male colleagues (49% vs. 32%; χ^2^ (1) = 4.544; *p* = 0.033). For the other three subcategories, no differences were observed between male and female therapists (*p* ≥ 0.112).

#### 3.2.7. Intensity of Psychotherapeutic Work

A final major category mentioned by *n* = 168 (77.4%) respondents subsumes statements about the intensity of psychotherapeutic work and comprises four subcategories, which are displayed in Figure 8. 

*N* = 70 (32.3%) respondents experienced high or even higher intensity in remote psychotherapy. This was explained by the fact that emotions can be expressed more openly in the remote setting, and difficult or shameful topics can be raised more easily, as respondent 73 described: “*The distance allowed some patients to be more open because there was less closeness and less shame*”. Respondent 193 put it this way: “*also an increased possibility to approach previously avoided contents from a distance*”. Patients were described as more disinhibited and open when they participated in sessions from the safety of their home environment. Respondent 79 observed, “*Some appreciated their familiar surroundings and were able to talk about more intimate topics*”. Themes activated by the pandemic also came into therapy and could be elaborated, which sometimes deepened the process, as respondent 62 reported: “*Patients perceive the switch* (to remote psychotherapy sessions) *as a form of caring (being concerned about them, making an effort, etc.), which sometimes also evokes memories, longings, deprivations,* etc. *regarding childhood”.* A greater density and thus intensity of the conversations was also described, here by respondent 196: “*With many patients, an increase in intensity was noticeable, the conversations were denser and more often led to a mutually satisfactory result*”.

In this context, *n* = 17 (7.8%) respondents also mentioned that the therapeutic work was more focused on topics or therapy goals. For example, respondent 177 commented, “*Condensed, rapid delving into all relevant topics of concern”.* Respondent 135 observed, “*For many patients, the work was even more to-the-point and focused on change”.* Respondents explained this as a result of increased concentration in remote contact and of the need for both parties to verbalize emotions more, as well as to focus attention on the available channels of perception and, in particular, on the spoken word.

However, respondents also made contrary observations. *N* = 128 (59%) respondents described that the intensity of therapeutic sessions decreased in remote psychotherapy, for example, because processes were disrupted by technical difficulties or because it was not possible to use the full range of interventions, or because patients did not engage emotionally and presented only everyday topics. More in-depth or biographical work was avoided, which was experienced by respondents as a flattening of the content. Respondent 6 summed this up with her statement, “*In some conversations, a kind of coffeehouse gossip atmosphere arose for a short time since otherwise you only talk on the phone with friends for such a long time”.* Respondent 124 also commented pointedly that patients remained in their “*comfort zone*”. Conversations were described as more rational and less emotional. 

COVID-19 as a topic and the issues the pandemic raised (coping with everyday life, fears, dealing with COVID-19 preventive measures, job loss, etc.) were the focus of remote psychotherapies. Respondent 94 commented, “*Conversations became more superficial. It became almost impossible to explore topics in depth. The topics were limited to current events and Covid measures, and the original goal of the therapy was neglected. The intensity of the conversations decreased a lot”.* Respondent 32 also noted that “*ongoing processes and reflections were interrupted”.* At the same time, it was emphasized several times that the engagement with daily events was not necessarily due to the switch to remote psychotherapy but was due to the crisis. “*It wasn’t the switch that changed the issues, it was the crisis that changed the issues”* (respondent 137).

In this context, *n* = 76 (35%) respondents stated that for them, the supportive function of therapy was the primary focus of remote contacts during the pandemic. This involved crisis intervention and counseling in the “*here & now*” (respondent 52) to relieve stress. Respondents described how they worked in a more supportive, resource-oriented and structuring way and were more directive and “*less exploratory*” (respondent 11). 

No gender differences became evident in the frequencies of all reports related to the subcategories relating to the intensity of therapeutic work (*p* ≥ 0.064).

## 4. Discussion

This study aimed to survey the changes experienced by Austrian psychotherapists when switching from face-to-face to remote psychotherapy in the first year of the COVID-19 pandemic. 

An important finding of the analysis is that neither therapeutic orientation nor years of professional experience had any influence on perceived changes when switching from face-to-face to remote psychotherapy or vice versa during the pandemic. This raises the assumption that differences between therapeutic orientations are sometimes given too much weight. As has already been shown in research on psychotherapy outcomes, different therapeutic orientations are similarly effective, and differences in effectiveness are due to factors other than therapeutic orientation [35,36]. 

This study further showed that working from home, especially the elimination of travel time to and from the office, allowed the surveyed therapists more flexibility in time management. This result is reflected in other studies [37,38,39]. Therapists with younger children, in particular, benefited from the greater flexibility. On the other hand, the elimination of time spent traveling also led to a loss of mental space for patients for reflective discussion before and after the session. This was also observed by Ahlström et al. [22]. In particular, the male therapists in our study reported missing the office as a therapeutic space. This could be because women found working from home more convenient, especially due to childcare responsibilities, and therefore did not miss face-to-face practice as much. In fact, another Austrian study showed that male psychotherapists treated more patients on average in face-to-face contact than female psychotherapists during the COVID-19 pandemic, which suggests that they continued working from their office or returned to their offices more rapidly [40].

Challenges mentioned by many respondents were the occurrence of technical problems and a reduced perception of sensory impressions. Technical problems were also reported in other studies from the same period [41,42]. As also noted by Jesser et al. [43] and Eichenberg et al. [44], therapists tried to compensate for the lack of non-verbal communication by focusing on other channels of perception. The respondents described this as exhausting and tiring, a finding that was also echoed by other authors [21,39,43]. The lack of a non-verbal level affected the therapeutic process. Respondents had difficulties in fine-tuning, found it harder to empathize, changed interventions and/or felt that an element of diagnostics was missing. Bayles et al. [45] argue that the quality of information is diminished in that therapeutic action is based on implicit and procedural non-verbal communication and that non-verbal information transmitted by the body in the setting at a distance is limited. Roesler [46] describes non-verbal information as essential in the process of mutual understanding. The loss or distortion of non-verbal elements has an impact on the patient’s emotional security [46]. The accompanying lack of affective nuance can emotionally weaken the therapist’s experience of working with the patient [41]. Respondents in our study also described a sense of loss of control in the remote setting. While they were able to choose their own space, they had no control over where their patients were or under what conditions they were attending. For many patients, finding an undisturbed space for confidential communication proved to be a challenge. This was also found by other authors [41,42,47]. The remote setting could also challenge patients to take more responsibility for themselves, which could be beneficial for patients with more moderate disorders. Simpson et al. [48], for example, described the “democratizing effect” of remote therapies, which enable patients to become more active in their own “territory”. Furthermore, Jesser et al. [43] worked out that in a setting at a distance, successes can also be experienced more independently of the therapist.

In many cases, psychotherapy via video conferencing also offered our respondents insight into the patient’s private environment and thus provided interesting additional information. Similarly, Jesser et al. [43] and Simpson et al. [48] described these insights as a unique opportunity to get a first-hand picture of the patients’ life circumstances described in the sessions. Respondents did, however, describe the distraction caused by other people or animals in the household as a challenge, a finding that is also consistent with findings from other studies [42]. Furthermore, there is the challenge of separating private and professional life, as Liberati et al. [49] and Shklarski et al. [50] also noted. A study by Békés et al. [51] concluded that therapists who faced more challenges when switching to the digital setting tended to be younger. This could be related to family responsibilities. Therapists with young children face challenges in creating a space where they have the opportunity to engage with their patients in a focused and empathetic way. However, we found no evidence in our study that therapist age had an impact on perceived challenges of the remote setting, or on other observed changes related to the remote setting. 

Respondents in our study, and female therapists in particular, mentioned the difficulties as well as the advantages of remote psychotherapy for certain diagnoses and patient groups. The observation that women seem to be more thoughtful about the difficulties and advantages of remote therapy for different patient groups and how to navigate this new setting could be attributed to women being more reflective and communicative in the study or in general [52]. Indeed, women provided, on average, 57% longer comments on free text questions vs. men. Respondents indicated that remote therapy, especially the setting via telephone, had proved helpful for patients with anxiety disorders. This finding is consistent with that of Jesser et al. [43], where respondents described that patients seemed more confident in remote treatment from their homes. Evidence from the research suggested the effectiveness of remote therapies for depression and/or anxiety disorders [10,12,13,14,15,53]. For the first time, according to our respondents, it was also possible for patients to address traumatic events in the setting at a distance. The remote treatment helped the patients to distance themselves from the events; furthermore, the setting was experienced as less fraught with feelings of shame. Previous study results already indicated the effectiveness of treating post-traumatic stress disorder (PTSD) in a distance setting [54,55] and described it as a viable alternative compared to the face-to-face setting [56]. By contrast, other respondents in our study considered the remote treatment of traumatized patients to be more difficult, as it did not enable patients to be supported as well, e.g., in the case of dissociative disorders. Other evidence from research can also be found for this [43,57]. In this context, we might need to consider that the COVID-19 pandemic itself and its associated constraints constituted a traumatic experience for some people. From the literature, we know that people were disposed of different resources protecting them against the traumatic experience of the pandemic. Killgore et al. [58] found that resilience was higher among people who, for example, maintained more social relationships, engaged in outdoor activities, and exercised more. Further research could examine whether patients experienced psychotherapy as helpful in coping with the pandemic and how patients with more or less resilience benefited differently from remote psychotherapies.

It was more difficult for our surveyed therapists to provide psychotherapeutic treatment for children in the setting at a distance. They could only accompany during play without actively participating or intervening. This is consistent with other research findings, which already pointed out that significant elements (e.g., creative opportunities) are lost or cannot be used in the remote treatment of children [50,59]. Instead, therapists focused more on their patients’ verbal communication, facial expressions, and tone of voice [60]. The ambiguity of the findings highlights the need for further research. How can the respective therapeutic methods be adapted to the remote treatment format [6], and are there possible contraindications for certain diagnoses or patient groups? 

Our results suggest a higher variability in the duration and frequency of sessions in remote therapy. This would suggest that, in addition to psychotherapeutic work, crisis intervention and counseling settings have been given more space in the respondents’ range of activities. Further research would be needed to determine to what extent the changed settings could be used for genuine therapeutic work or whether the focus of the work had shifted. 

A significant issue indicated by our research is the abandonment or restriction of the use of therapeutic interventions in distance therapy. This was also observed by Cantone et al. [61]. Notermans et al. [62] found that interventions that intend to activate intense or aversive feelings are avoided in the setting of remote therapy. Probst et al. [26] also concluded that therapeutic interventions are considered more typical for face-to-face psychotherapy than for psychotherapy at a distance. This could be explained by the fact that in training, the use of therapeutic interventions has so far been taught exclusively in the context of face-to-face psychotherapy [26]. Further research is needed to determine whether genuine interventions can be adapted for remote therapy. Therapists may have been uncertain about using certain interventions in the remote setting due to a lack of experience. This could be counteracted by offering training on remote treatment that is rooted in education and training contexts.

We noticed a great ambivalence in our respondents’ answers regarding the relational experiences and intensity experienced in the remote sessions. The continuation of therapy during the period of restrictions on outdoor activities was described by respondents as having a confidence-building and relationship-strengthening effect. Female therapists, in particular, described a strengthening of relationships. Arguing from a sociological perspective, this finding could be explained by women taking on more nurturing roles in society [63]. Research has shown that women shouldered much of the increased demands of housework and childcare during the pandemic [64]. It could be hypothesized that women are also more likely to take a nurturing role in therapy. Indeed, psychotherapy research has shown that female psychotherapists are more loyal, more optimistic, and less critical than their male colleagues and also more able to put their own person in the background [65]. While male therapists tend to use more confrontational techniques, female therapists intervene more empathically [66]. As a result, women may also be more likely to perceive the gratitude of their patients, which was particularly important during the pandemic. Huscsava et al. [42] and Bouchard et al. [11] also came to the conclusion that therapeutic relationships were strengthened by the continuation of psychotherapy in a remote setting during the pandemic. One narrative review already published in 2014 was able to show that for patients, the therapeutic relationship in psychotherapy via videoconferencing does not differ from the face-to-face setting [67]. Stoll et al. [68] also rated the therapeutic relationship in the online setting as equal or even better compared to the face-to-face setting. Some of the respondents also perceived a high or higher intensity. Emotions were expressed more openly; furthermore, difficult or embarrassing topics could be addressed more easily. A study by Stefan et al. [21] already provided indications that patients can open up more easily about embarrassing topics over the telephone. It seems that this effect is not limited to psychotherapy via telephone, as patients also felt more confident and less intimidated to talk openly about their emotional state and problems in the setting of videoconferencing [67]. In this context, Russell [69] pointed to the disinhibitory effect in online settings, which leads to some patients opening up more emotionally in video conferencing or telephone settings. Furthermore, Roesler [46] described the intensification effect, which often occurs in the context of virtual interaction. In this situation, information that is only transmitted in a restricted way is completed through the use of fantasy in an imaginative process that also includes the processes of projection and transference [46].

On the other hand, some respondents described a decrease in the intensity of the therapeutic sessions in the setting of distance therapy. Other authors also reported that the therapeutic work became more superficial in terms of content [20,22,42] and that the topics were increasingly oriented toward the patients’ everyday life [42]. Some of the respondents also perceived less closeness in the psychotherapeutic relationship in remote therapy and/or experienced the establishment of a sustainable psychotherapeutic relationship as challenging. Psychotherapists interviewed in the study by Stefan et al. [21] also described the therapeutic relationship as more superficial. The respondents in our study also reported limited possibilities of being able to emotionally support the patient in the setting at a distance. Therapists, according to Germain et al. [70], may feel that they can only support their patients in a limited way (e.g., because they cannot offer a handkerchief). Huscsava et al. [42] concluded that therapists feel more insecure in the event of a crisis due to limited options for taking action. 

The hypothesis put forward by Roesler [46] is that using technological means to interact psychotherapeutically leads to a fundamental change in interpersonal encounters, the intrinsic rules and consequences of which are still not understood sufficiently well. Given the ambivalence and ambiguity of the empirical findings found in various studies [43,51], it is clear that further research and, in particular, observational studies are needed to better understand interaction in the remote setting, especially in the absence of pandemic conditions. 

There are several limitations to this study. Firstly, it is a non-randomized study with some confounding factors that might influence the results (e.g., experiences of tele-psychotherapy mainly relate to the time during the COVID-19 restrictions). Secondly, the cross-sectional design did not allow for obtaining therapists’ experiences session by session, which in turn could lead to recall bias in the retrospective assessment of the change experienced when making the switch to remote therapy. Thirdly, only psychotherapists who had entered a valid email address in the Austrian list of psychotherapists were reached. Fourth, the survey was conducted online, which could lead to the higher participation of therapists with a higher preference for psychotherapy via videoconferencing. Fifth, it may not be possible to generalize the results to other countries since e-mental health services already have a long tradition in other countries, and therapists’ attitudes and experiences may therefore differ. Finally, it would be interesting to investigate possible changes in therapists’ attitudes toward the setting at a distance over time.

## 5. Conclusions

As a result of the COVID-19 pandemic, the forced and abrupt change in psychotherapeutic treatment format from face-to-face settings to remote psychotherapy faced psychotherapists with unique and complex challenges [50]. Our study showed that remote psychotherapy can be an option to ensure continuity in case of a crisis. Furthermore, the setting offers spatial and temporal flexibility, which means that appointments can be offered more quickly in case of the need for a higher frequency of sessions or in case of crises. Our study indicates that for some disorders (e.g., anxiety disorders), treatment at a distance does have benefits. Further clinical studies are needed to identify how these patients benefit from distance treatment.

At the same time, it was found that the change of setting led to feelings of insecurity on the part of the therapists and that the range of therapeutic interventions was not fully utilized. This underscores the relevance of further research on how the therapeutic methodology can be adapted to the remote setting and for which patients there might be a contraindication. Because remote treatment has not been included in the process of professionalization so far, we see a need to expand the training and further education offered to therapists accordingly. At any rate, the pandemic situation has shown that to fulfill the duty of care toward patients, new ways are needed to ensure psychotherapeutic care [48]. Treatment at a distance could constitute an alternative to counteract the already existing underprovision of psychotherapeutic care in Austria.

## Figures and Tables

**Figure 1 life-13-00360-f001:**
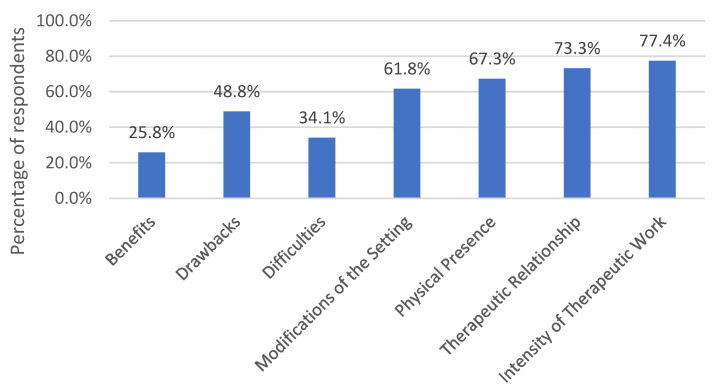
Main categories of the qualitative content analysis and the percentage of respondents reporting one or more experiences in each of the main categories. The percentages of the main categories may differ from the sum of the percentages in the individual subcategories (Figure 2, Figure 3, Figure 4, Figure 5, Figure 6, Figure 7 and Figure 8) because it may be that a respondent reported experiences in several subcategories (e.g., technical problems and distraction) within one main category (e.g., drawbacks) and thus appears in each of these subcategories but is only counted once per main category.

**Figure 2 life-13-00360-f002:**
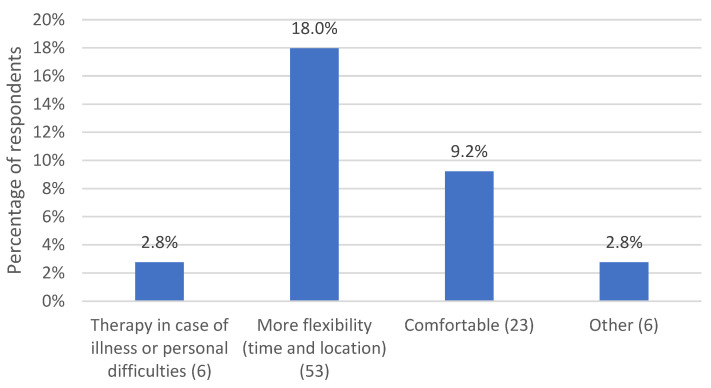
Percentage of respondents who experienced various benefits of remote psychotherapy. The number in parentheses after the subcategory name indicates the number of coded text passages.

**Figure 3 life-13-00360-f003:**
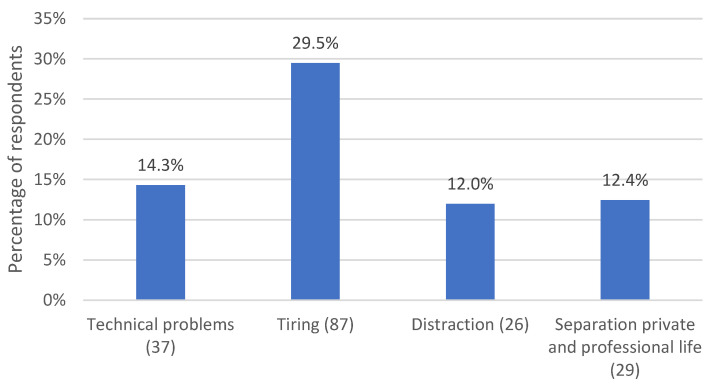
Percentage of respondents who experienced various drawbacks of remote psychotherapy. The number in parentheses after the subcategory name indicates the number of coded text passages.

**Figure 4 life-13-00360-f004:**
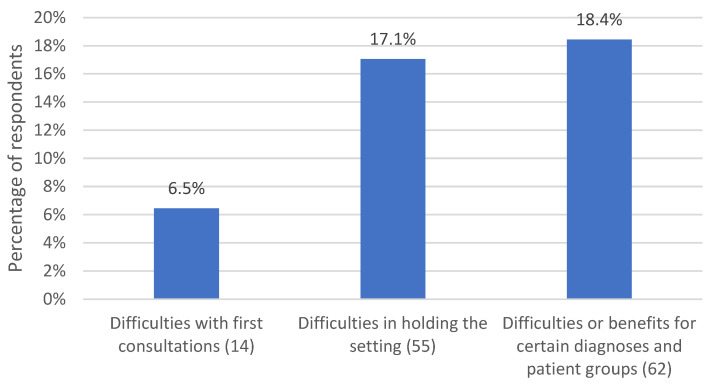
Percentage of respondents who experienced various difficulties of remote psychotherapy. The number in parentheses after the subcategory name indicates the number of coded text passages.

**Figure 5 life-13-00360-f005:**
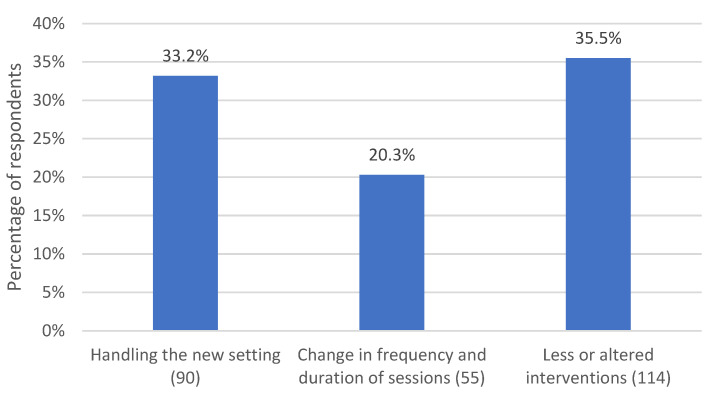
Percentage of respondents who reported modifications of the setting. The number in parentheses after the subcategory name indicates the number of coded text passages.

**Figure 6 life-13-00360-f006:**
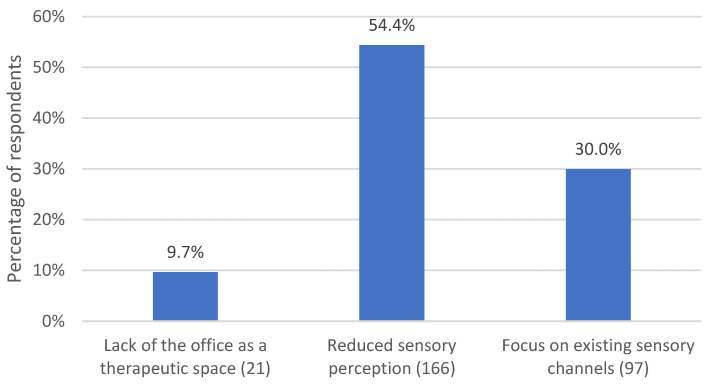
Percentage of respondents who commented on diverse experiences with the lack of physical presence. The number in parentheses after the subcategory name indicates the number of coded text passages.

**Figure 7 life-13-00360-f007:**
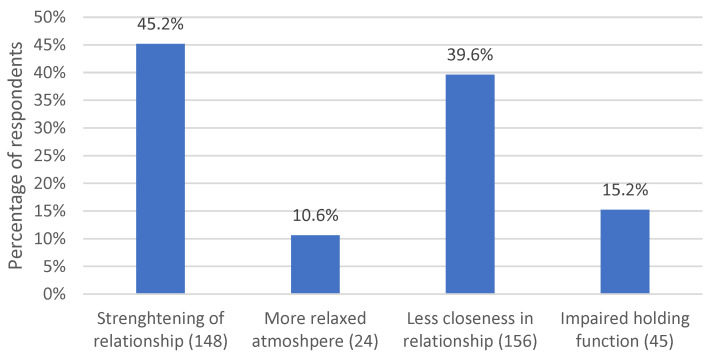
Percentage of respondents who commented on aspects relating to the quality and intensity of the psychotherapeutic relationship. The number in parentheses after the subcategory name indicates the number of coded text passages.

**Figure 8 life-13-00360-f008:**
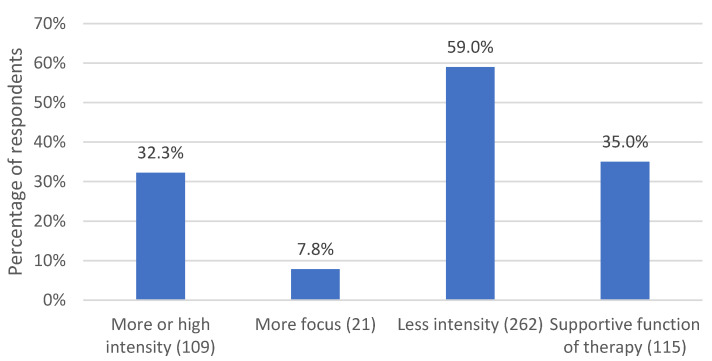
Percentage of respondents who commented on their perceptions regarding the intensity of psychotherapeutic work. The number in parentheses after the subcategory name indicates the number of coded text passages.

**Table 1 life-13-00360-t001:** Main categories of the qualitative content analysis by gender.

Main Category	Female (*n* = 167)	Male (*n* = 50)	Statistics
Benefits, % (N)	25.1% (42)	28.0% (14)	χ^2^ (1) = 0.163;
			*p* = 0.686
Drawbacks, % (N)	48.5% (81)	48.0% (24)	χ^2^ (1) = 0.004;
			*p* = 0.950
Difficulties, % (N)	37.7% (63)	20.0% (10)	χ^2^ (1) = 5.415; *p* = 0.020
Modifications of the Setting, % (N)	64.7% (108)	50.0% (25)	χ^2^ (1) = 3.491; *p* = 0.062
Physical Presence, % (N)	68.3% (114)	64.0% (32)	χ^2^ (1) = 0.318; *p* = 0.573
Therapeutic Relationship, % (N)	79.0% (132)	52.0% (26)	χ^2^ (1) = 14.214; *p* < 0.001
Intensity of Therapeutic Work, % (N)	79.0% (132)	70.0% (35)	χ^2^ (1) = 1.774; *p* = 0.183

## Data Availability

The raw data supporting the conclusions of this article will be made available by the authors upon reasonable request after signing a confidentiality agreement.

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
