# Peer review of "Remote Psychotherapy during the COVID-19 Pandemic: A Mixed-Methods Study on the Changes Experienced by Austrian Psychotherapists"

_life, 2023, doi:10.3390/life13020360_

Round 1

Reviewer 1 Report

The manuscript entitled " Remote Psychotherapy During the COVID-19 Pandemic: A 2 Qualitative Study Using Content Analysis on the Changes 3 Experienced by Austrian Psychotherapists" explore the perceptions on the changes experienced by Austrian therapists switching to remote psychotherapy during the pandemic.

In the introduction section, in objectives, reference is made to methodological aspects (as: “An exploratory research approach was adopted…”; which could be more appropriate to integrate in the methods section). Reference is made to a primary (or general) objective, but secondary (specific) objectives and research hypotheses are not specified. It might be appropriate to organize into general and specific objectives.

At the methodological level, section 2.1 refers to procedural aspects, so it should be reflected in the title of the section. In the participants section, the inclusion and exclusion criteria considered to participate in the study should be specified. The estimated average time for participants to respond to the online survey could also be indicated (researchers' estimate, average time spent by participants if they registered, etc.). In the Data Analysis section, it could be further specified that it is: “conventional approach to qualitative content analysis”.

It would be appropriate to carry out a statistical analysis between qualitative variables (for example, using Chi square) of interest (experienced benefits, experienced drawbacks of remote psychotherapy, etc.) and sociodemographic-professional (gender, psychotherapeutic orientation, etc.) variables to have a more specific knowledge at a qualitative-statistical level of the possible relationships that may exist between them. To facilitate the reading of the results section, it should focus on describing only the main findings, including the opinions of the participants as supplementary material in relation to each category.

The discussion section should consider aspects to include in relation to the possible relationships between qualitative variables that could be derived from the proposed statistical analysis.

Author Response

Dear reviewer,

Thank you for the time and effort you took to review our paper!

As you suggested, we adapted the last part of the introduction section. We moved the information addressing the research approach to the methods section and included specific secondary objectives.

In the methods section, we have changed the heading 2.1. so that the procedural character is more strongly expressed. Furthermore, we provided additional information addressing inclusion/exclusion criteria and regarding the estimated time for participants to respond to the online survey. Unfortunately, RedCap does not provide any information on how long participants actually took to complete the questionnaire in its entirety. We have also added a sentence to explain the conventional approach to qualitative content analysis in a little more detail.

You also recommended to include statistical analyses between categories resulting from the qualitative content analysis and sociodemographic and professional variables. We followed your recommendation and conducted a mixed-methods analysis to obtain information on potential differences in the frequency of main categories reported by psychotherapists in terms of sociodemographic (age, gender) and professional characteristics (professional experience, psychotherapeutic orientation). Consequently, we have added a paragraph on the statistical analysis in the methods section and expanded the presentation of results and discussion accordingly.

Another recommendation in your report was to streamline the results and include the participants quotes as supplementary material. We have carefully considered your advice but have come to the conclusion that we would like to retain the current form of presentation. We would like to justify our decision by referring to the APA Journal Article Reporting Standards for Qualitative Research in Psychology, which we followed in the presentation of our research. There it is stated that “Qualitative approaches to inquiry may utilize distinct styles of reporting that may still be unfamiliar to many psychologists and social scientists (Sandelowski & Leeman,2012). These can include a narrative style of reporting, in which the research endeavor is presented as a story. (…) Results sections also tend to be lengthy because the methodological integrity of qualitative methods is enhanced within a demonstrative rhetoric in which authors show how they moved within the analysis from their raw data to develop their findings.” We kindly ask you to re-evaluate our paper against this background and would be pleased if you shared our assessment to leave the results in their current form.

Reviewer 2 Report

the paper is very interesting and well done. It can give a good contribution from a scientific point of view. I suggest some changes. For example, it would be better to start a period with a number written in letters (see line 139) and not in numbers. Also, graphs should report only one type of measurement, that is, either quantities or percentages and not both. Some figure captioning contains information better suited for the text of the article, and not for a figure caption (e.g., figure 1). 

Finally, the authors, in discussions, claim "remote therapy, especially the setting via telephone, had proved helpful for patients with anxiety disorders," and also "it was also possible for patients to address traumatic events in the setting at a distance." In this respect, I suggest that a more in-depth analysis should be done for the purpose of better interpretation of the results especially with respect to the resilient resources used by people under conditions of trauma and in times of pandemic. I suggest the following studies:

Renati, R., Bonfiglio N.S., Rollo D. (2023). Italian university students' resilience during the covid-19 lock-2 down. A structural equation model about the relationship be-3 tween resilience, emotion regulation and well-being.

Killgore, W. D., Taylor, E. C., Cloonan, S. A., & Dailey, N. S. (2020). Psychological resilience during the COVID-19 lockdown. Psychiatry research, 291, 113216.

Author Response

Dear reviewer,

Thank you very much for the time and effort of reviewing our paper and providing valuable input to help us improve the paper further. As you suggested, we have changed the figures to show only percentages. We also shortened the figure captions and included information in the main body of the text. With regard to your comment “it would be better to start a period with a number written in letters (see line 139) and not in numbers”, we were unsure what you meant. Line 139 is a chapter heading in our document and the formatting for this is specified by the journal.

Thank you as well for the papers on resilience during the pandemic that you suggested mentioning in the discussion. We have tried retrieving the first paper you mentioned, but couldn`t find it in various databases. However, we have studied the second paper and have added a paragraph in the discussion. We hope that you can now endorse a publication.

Reviewer 3 Report

Very good paper. This research examines changes during the COVID, and how it was perceived in therapy. It's very relevant to this field, not many studies on psychotherapy via tele model. The perception of the therapist and first-hand experience were added to the subject area compared with other published material. Regarding the methodology, maybe mention some Qualitative study reporting guidelines if not done yet. The conclusions are consistent with the evidence and arguments presented and address the main question posed.

Author Response

Dear reviewer,

Thank you very much for the time and effort of reviewing our paper. We appreciate your feedback and your positive opinion. As you suggested, we added a reference to the APA Journal Article Reporting Standards for Qualitative Research in Psychology, we adhered to. We hope that you can endorse a publication.

Round 2

Reviewer 1 Report

The different modifications requested to the article have been adequately included. In my opinion, the document would be adequate in its current format. Thanks for their corrections.